# Rural–Urban Differences in Dietary Behavior and Obesity: Results of the Riskesdas Study in 10–18-Year-Old Indonesian Children and Adolescents

**DOI:** 10.3390/nu11112813

**Published:** 2019-11-18

**Authors:** Esti Nurwanti, Hamam Hadi, Jung-Su Chang, Jane C.-J. Chao, Bunga Astria Paramashanti, Joel Gittelsohn, Chyi-Huey Bai

**Affiliations:** 1International Master/Ph.D. Program in Medicine, College of Medicine, Taipei Medical University, Taipei 11031, Taiwan; estinurwanti3@gmail.com; 2Department of Nutrition, Faculty of Health Science, Universitas Alma Ata, Yogyakarta 55183, Indonesia; hamamhadi99@gmail.com (H.H.); pshanti.bunga@gmail.com (B.A.P.); 3Alma Ata Center for Healthy Life and Foods (ACHEAF), Universitas Alma Ata, Yogyakarta 55183, Indonesia; 4School of Nutrition and Health Sciences, College of Nutrition, Taipei Medical University, Taipei 11031, Taiwan; susanchang@tmu.edu.tw (J.-S.C.); chenjui@tmu.edu.tw (J.C.-J.C.); 5Graduate Institute of Metabolism and Obesity Sciences, College of Nutrition, Taipei Medical University, Taipei 11031, Taiwan; 6Master Program in Global Health and Development, College of Public Health, Taipei Medical University, Taipei 11031, Taiwan; 7Nutrition Research Center, Taipei Medical University Hospital, Taipei 11031, Taiwan; 8Sydney School of Public Health, The University of Sydney, Sydney, NSW 2006, Australia; 9Center for Human Nutrition, Department of International Health, Bloomberg School of Public Health, The Johns Hopkins University, 615 North Wolf Street, Baltimore, MD 21205-2179, USA; jgittel1@jhu.edu; 10Department of Public Health, College of Medicine, Taipei Medical University, Taipei 11031, Taiwan; 11School of Public Health, College of Public Health, Taipei Medical University, Taipei 11031, Taiwan

**Keywords:** adolescents, overweight, obesity, urban, rural, diet, Indonesia

## Abstract

Obesity has become a significant problem for developing countries, including Indonesia. High duration of sedentary activity and high intake of unhealthy foods were associated with high risk of overweight and obesity. The objective of this study was to compare the distributions of sedentary activity and dietary behavior with overweight/obesity risks between urban and rural areas among children and adolescents aged 10–18 years in Indonesia. This is a cross-sectional study. Data from a national survey in 33 Indonesian provinces (Basic Health Research /Riskesdas 2013) were analyzed. Multiple logistic regression models were used to calculate the odds ratio (OR) adjusted with all variables, such as age, gender, residency, education level, physical activity, and food intake. An urban–rural residence difference was found in the factors related to obesity. Daily caffeinated soft drinks and energy drinks consumption (OR = 1.12, 95% CI: 1.01–1.23) were related to risk of overweight and obesity in urban areas. Daily grilled foods (OR = 1.32, 95% CI: 1.22–1.42) and salty food (OR = 1.09, 95% CI: 1.04–1.15) consumption were significantly associated with obesity in rural areas but not in urban areas. Furthermore, sedentary activity was correlated with overweight and obesity among those who lived in urban and rural areas. Our findings suggest that education, environmental, and policy interventions may need to specifically target urban settings, where access is high to a wide range of processed and traditional high-sugar, high-fat snack foods and beverages.

## 1. Introduction

Overweight and obesity among children and young adults have become emerging issues in every region of the world, including in low- and middle-income countries (LMICs) [1,2,3,4,5,6]. In the developing world, the prevalence of overweight and obesity was 28.8% and 2.3%–12.0%, respectively [4], similar to rates in higher-income countries [5]. Although there was a wide variation in overweight and obesity rates between countries, over the last 33 years, there has been no significant improvement in obesity reduction [7]. Additionally, in developing countries, the burden has doubled as obesity prevalence has risen alongside a persistent burden of undernutrition [1,8]. In Indonesia, overweight and obesity have become public health problems in parallel with undernutrition. The prevalence of adult overweight and obesity have increased from 8.6% and 10.5% in 2007 to 13.6% and 21.8% in 2018 [9], respectively. Among children aged 5–12 years, the proportion of overweight remained unchanged between 2013 and 2018 (10.8%), whereas the obesity prevalence slightly increased from 8.8% to 9.2% during these years [9,10]. Furthermore, central obese prevalence among >18 years population increased dramatically from 18.8% in 2007 to 31.0% in 2018 [9].

The determinants of overweight and obesity in LMICs are not well understood and could be driven by the changes in dietary behaviors and physical activity, as well as environmental factors, genetics and epigenetics, and stress. It has been suggested that diets have become higher in calories, sugar, fat, and salt, whereas physical activity level has reduced worldwide [1,7]. Maternal and household factors in early life may also bring demographic and socioeconomic changes, which can lead to higher body mass index (BMI) in later life [4,11].

In LMICs, the prevalence of overweight and obesity is typically higher in urban areas while underweight is typically higher in rural areas [12,13,14]. Children who live in urban areas tend to have a higher risk of overweight and obesity due to unhealthy dietary behaviors, such as high intake of sugar-sweetened beverages intake [15,16]. The dietary behaviors in urban areas are also more diverse compared to rural areas [17]. In contrast, among rural and semi-urban residents, salt is added almost regularly to cooking foods, and salted foods such as fish and meat are eaten frequently [18]. Subsequently, the association between dietary behavior and risk of overweight/obesity remains unclear after being stratified by rural and urban locale, and after adjustment for other factors.

To our knowledge, no study has reported differences in rates of overweight and obesity between urban- and rural-dwelling 10–18-year-old children and adolescents in Indonesia, along with the associated risk factors regarding dietary behavior. Therefore, our aim is to analyze the determinants of dietary behavior in children and adolescents in Indonesia. The second purpose of this study is to (1) examine the prevalence of overweight, and obesity; (2) explore the percentages of sedentary activity and dietary behavior; and (3) compare sedentary activity and dietary behavior with overweight/obese risk between urban and rural areas among children and adolescents aged 10–18 years in Indonesia. Given the knowledge that overweight and obesity proportions are higher in urban compared to rural areas, the findings of this study may contribute to the development of targeted overweight and obesity reduction programs in Indonesia based on rural–urban-specific causes.

## 2. Materials and Methods

### 2.1. Study Population and Design

We analyzed data from a cross-sectional, nationally representative survey (Indonesia Basic Health Research 2013/Riskesdas 2013/Riset Kesehatan Dasar 2013) conducted by the National Institute of Health Research Development (NIHRD), Ministry of Health, Indonesia in 2013. The methodology and the detailed protocol used are described elsewhere [10]. Briefly, the selection of participants was carried out by a two-stage stratified cluster sample drawn from across the country, including 33 provinces and 497 municipalities/districts in Indonesian. During a home visit, trained interviewers collected primary data on children and adolescents’ characteristics and other measurements. Well-trained interviewers collected anthropometric measurements (height and weight) using a standardized protocol. The inclusion criteria in this study were: (1) Children and adolescents aged 10–18 years old and (2) no missing data. The exclusion criteria were children and adolescents with missing data, such as weight, height, duration of sedentary behavior, and fruit vegetable intake. Data from 185,631 children and adolescents were available, however, there were 29,986 participants with missing data excluded from this study. Finally, 155,645 adolescents (83.85%) aged 10 to 18 years were included in this study.

### 2.2. Ethical Considerations

The Health Research Ethics Committee—University of Indonesia, Hasanuddin University, Airlangga University, and the National Institute of Health Research and Development (NIHRD), Ministry of Health Republic of Indonesia, reviewed and approved the protocol, design, data, and survey questionnaires (LB.02.01/5.2/KE.006/2013). No further ethical clearance was required for the secondary analysis of published public data. All of the participants were asked for their consent and signed the informed consent form in the survey.

### 2.3. Socio-Demographic and Dietary Behavior Questionnaire

Age, residency, gender, and education level were collected using a validated questionnaire and interview performed on children and adolescents [10,19]. A simple questionnaire and food cards were used to ask the frequency of consumption of 9 unhealthy food items, including sugar-sweetened beverages and foods, coffee and caffeinated soft drinks and energy drinks, fatty and fried foods, refined carbohydrates, salty foods, instant noodles, preserved meats, and grilled foods. Furthermore, this simple questionnaire also asked about the portions of fruits and vegetables consumed per day. The frequency of dietary behavior was evaluated by the question “how often do you consume?” with six possible answers: “>1 time/day”; “1 time/day”; “3–6 times/week”; “1–2 times/week”; “≤3 times/month”; and “never”. Then, in this study, dietary behavior was categorized into <1 time/day and ≥1 time/day.

The following is a description of unhealthy foods included in this study: (a) Refined carbohydrates included prepared flour products with added sugar, such as flavored bread; (b) fatty fried foods included fatty foods such as fatty meats, oxtail soup, fried foods, foods containing coconut milk and margarine, and high-cholesterol foods such as innards (intestines, tripe), eggs, and shrimps; and (c) sweet foods and drinks include high-sugar foods and beverages with additional natural sugar, such as cakes, canned fruit, artificial juice, soft drinks, syrups, and sweet tea. Soft drinks and sweetened drinks with zero calories and low sugar and diet drinks were excluded; (d) coffee; (e) caffeinated soft drink and energy drink, included caffeinated drinks except coffee, such as caffeinated soft drinks, and energy drink; (f) salty foods: Foods that are predominantly salty, like salted fish, fish *pindang*, salted eggs, salty snacks, foods containing shrimp paste, soy sauce, and sauce; (g) grilled foods: Foods that are processed by burning on fire directly, for example satay, grilled chicken, rolled goat, grilled fish; and (h) preserved meat: Beef or pork through processing and preservation use additives (nitrate). Preserved meat usually has red characteristics, for example, corned beef, sausages, meat burger, and smoked meat.

The definition of fruits and vegetables are all types of fruits and vegetables in Indonesia, both local and imported. In addition, the frequency of fruits and vegetable was categorized into <5 portion/day and ≥5portion/day [10].

### 2.4. Anthropometric Data

Height was measured by well-trained interviewers to the nearest 0.1 cm by a multi-function brand Stadiometer. A Camry digital weight scale with a capacity of 150 kg was used to measure body weight. Before use, the weight scale was calibrated daily. Body mass index (BMI) was calculated as weight (kg) divided by squared height (m^2^). It was categorized using gender-specific of body mass index (BMI) per age according to CDC criteria: Underweight (less than 5th percentile), normal or healthy weight (5th–<85th percentile), overweight (85th–<95th percentile), and obese (≥95th percentile). Body mass index (BMI) per age based on the gender of the Indonesian children and adolescents are summarized in Appendix A. Furthermore, adolescents were categorized as underweight, normal, overweight, or obese based on this national reference.

### 2.5. Physical Activity Questionnaire

An interview questionnaire was used to record types of sedentary activity. Participants were asked to recall all activities that started from waking up in the morning to going to sleep at night. Participants were asked how many hours or minutes they typically spent at work or home, sitting for transportation (including waiting for transport), sitting or lying down to watch TV or play electronic games, or sitting or lying down to carry out other activities (such as speaking, reading, etc.), not including sleeping. Total sedentary activity time (hours/day) was calculated based on time spent in all the above listed sedentary activity subcategories [10] and sedentary activities were coded as <2 h/day, 2–4 h/day, >4–6 h/day, >6–8 h/day, and >2 h/day.

### 2.6. Statistical Analysis

First, we examined the prevalence of overweight and obesity by using percentage. Second, *chi-square* tests were conducted to explore residence differences in overweight and obesity status, dietary behaviors, and other variables. Third, the odds ratio (OR) and 95% confidence intervals (CIs) for the association of factors related to overweight and obesity were estimated using logistic regression analysis. We controlled for all variables, such as residency, gender, age, education level, physical activity, and food intake in the regression analysis. In addition, we also stratified the models by gender and residency to investigate potential gender and residency difference in these associations. The data were analyzed using the SAS Version 9.4 (SAS Institute Inc., Cary, NC, USA); *p* < 0.05 was deemed statistically significant.

## 3. Results

### 3.1. Prevalence of Obesity in Rural and Urban Areas

The prevalence of overweight and obesity among children and adolescent was higher in urban areas (17%) than in rural areas (13.5%). Moreover, the prevalence of obesity in urban areas was higher in every age category of children and adolescents (Figure 1). 

After stratifying by gender, boys (15.2%) had a slightly higher prevalence of overweight and obesity than girls (15.1%). In addition, among boys and girls, the prevalence of overweight and obesity was still higher in urban areas. Meanwhile, in urban areas, boys had a higher prevalence of overweight and obesity than girls (Figure 2).

### 3.2. Sedentary Activity and Dietary Behaviour in Rural and Urban Areas

Sedentary activity was more prevalent in urban areas than in rural areas. Urban areas had higher mean hours of sedentary activity per day than in rural areas (Figure 3). However, sedentary activity was associated with a higher risk of overweight and obesity both in urban (OR = 1.09, 95% CI: 1.02–1.16) and in rural (OR = 1.10, 95% CI: 1.04–1.17) areas (Table 1).

We found significant differences in dietary behaviors among children depending on locale (Table 2). There was a significant association between sugar-sweetened beverages, coffee, caffeinated soft drink and energy drink, fatty fried foods, refined carbohydrate, preserved meat, and grilled foods consumption with locale (rural and urban). In this study, we also reported that there was no significant difference in intakes of salty foods and instant noodles between rural and urban areas. However, only among boys, lower intake of vegetables and fruits consumption was significantly higher in rural than urban areas.

Furthermore, children and adolescents who lived in rural areas had a higher percentage of coffee and grilled foods consumption, while those in urban areas had a higher percentage of sugar-sweetened beverages, caffeinated soft drink and energy drink, fatty fried foods, refined carbohydrate, and preserved meat consumption.

### 3.3. Overweight and Obesity Risk in Rural and Urban Areas

There was a significant difference found in the risk of overweight and obesity between urban and rural areas. In a multivariate logistic regression model analysis, after controlling for all variables, children and adolescents living in urban areas (OR = 1.32, 95% CI: 1.28–1.35), with sedentary activity (OR = 1.10, 95% CI: 1.05–1.14), consuming daily caffeinated soft drinks and energy drinks consumption (OR = 1.11, 95% CI: 1.03–1.19), and daily grilled foods intakes (OR = 1.19, 95% CI: 1.12–1.26) were all significantly associated with a higher risk of overweight and obesity. In contrast, low education level (OR = 0.92, 95% CI: 0.89–0.96) and low intake of daily fruits and vegetable consumption (<5 portions/day) (OR = 0.92, 95% CI: 0.88–0.98) significantly reduced overweight and obesity risk in rural and urban areas (Table 1).

Sedentary activity was associated with overweight and obesity risk, not only in rural areas, but also in urban areas. In rural areas, there was no significant association of gender and age with overweight and obesity risk, but, there was a significant association between low education level (OR = 0.93, 95%CI: 0.87–0.98), sedentary activity 2–4 h/day (OR = 1.10, 95% CI: 1.03–1.17), daily salty foods consumption (OR = 1.09, 95% CI: 1.04–1.15), and daily grilled foods consumption (OR = 1.31, 95% CI: 1.22–1.42) with overweight and obesity risks. Furthermore, in urban areas, being a boy (OR = 1.05, 95%CI: 1.01–1.10), of younger age (OR = 1.10, 95%CI: 1.04–1.17), with sedentary activity 2–4 h/day (OR = 1.11, 95%CI: 1.04–1.19) and frequently consuming caffeinated soft drink and energy drink (OR = 1.12, 95%CI: 1.01–1.23) was significantly associated with overweight and obesity risk.

## 4. Discussion

This is the first nationally representative study in Indonesia exploring the differences between weight status, and dietary and physical activity behaviors among children and adolescents in urban and rural areas. We found that urban residence was associated with greater obesity risk. Studies in developing countries, such as Vietnam, Malaysia, Indonesia, and Thailand, have also shown that the prevalence of overweight and obesity is higher in urban areas [13,14,20,21], agreeing with our findings. In developing countries, the economic growth resulting from the nutrition transition affects individuals and families, increasing their access to processed and fast food [22,23,24,25].

Urbanization is associated with a dramatic reduction in physical activity and changes in dietary behavior [26]. Rural-to-urban migration may change dietary behaviors from consuming main staples, such as grains and legumes, to regularly consuming energy-dense and animal-source foods. The pattern increases with wealth in urban areas [27]. On the contrary, among lower-wealth families, poor diet and limited access for physical activity may increase their vulnerability of becoming overweight or obese [28]. However, living in urban areas may also present certain challenges related to physical activity due to the residential and public transport density, and the availability of public parks [29]. Urban areas may be considered obesogenic environments [30], with advertisements prominently delivering messages to increase fast foods, processed foods, and beverages consumption that are high in carbohydrates, sugar, fat, and salt [31].

There was no association between sex and overweight or obesity when being analyzed for both rural and urban. Nonetheless, urban boys were more likely to become overweight and obese than urban girls. In this study, boys tended to become more obese than girls likely due to having a higher intake of sugar-sweetened beverages and caffeinated soft drink and energy drink than girls. In terms of sedentary activity, both in boys and girls in urban or rural, sedentary activity has a significant association with overweight and obesity. However, a previous study found that girls tend to be less active than boys at all ages [32,33,34]. Fear of injury, sense of safety, and reliance on others for everyday mobility could be the reason that girls participated less than boys [35]. On the other hand, there was no difference between boys and girls living in rural areas in regard to overweight and obesity risk. A study in Portugal reported that boys who were living in urban areas participated in various sports activities compared to those living in rural areas [36]. It seems that rural children and adolescents of both sexes have more difficulty in accessing sport or fitness centers and any other facilities for performing physical activity compared to urban-living children and adolescents.

A previous study reported that children and adolescents consuming any sweet drink or soft drink was >70% and the proportion remained steady across time [37]. A cross-sectional study that included 75 countries found that soft drink consumption is significantly associated with overweight and obesity, especially in low- and middle-income countries [38]. In particular, higher soda drink consumption was associated with weight gain, contributing approximately 8–9% of child and adult total energy intakes, thus leading to obesity [39]. In this study, daily caffeinated beverages consumption, which includes energy drinks and caffeinated soft drinks, are also a risk factor for overweight and obesity, particularly in urban areas, a finding in line with other studies [40]. While caffeine is the key ingredient in most energy drinks, most products available on the market also contain a high amount of glucose or artificial sweetener [41], which may later contribute to higher energy intake. Moreover, the consumption of energy drinks presumably leads to higher carbohydrate oxidation and lower lipid oxidation [42]. Regarding soft drinks, besides a high level of added sugar, poor satiety and insufficient total energy compensation may influence weight gain [39]. Finally, soft drink consumption can be used as an indicator of poor diet as it is linked to increased calorie intake and may displace nutritious food [43]. For regular caffeinated sugar-sweetened beverage consumption, caffeine content was found to be additive, thus promoting repeated consumption of sugar-sweetened beverages themselves, thus leading to weight gain [44]. Children and adolescents in urban areas may be better able to find caffeinated beverages than in rural area. A policy to limit caffeinated beverages consumption in urban areas is needed, particularly targeting regulations in the advertisement, sale, and location of food retailers. School- and family-level supports are required to help children and adolescents make healthy food choices.

In rural Indonesia, participants with higher daily salty foods and grilled foods consumptions had higher risk factors for overweight and obesity. Salty snacks were associated with higher increases in mean daily energy for children [45]. A previous study showed that long-term consumption of a high salt diet is associated with increased frequency of obesity [46]. Longitudinal studies have also shown that high salt consumption predicts the development of obesity, even when total energy intake or consumption of sugary drinks has been controlled [47]. Furthermore, a previous study also found that regular servings of energy-dense snack foods, which have high sodium, had a direct relationship with body mass index (BMI) z-score among adolescents [48].

In this study, grilled foods, including grilled beef, chicken, goat, and fish, were related to meat consumption. A previous prospective study found that total consumption of meat was positively associated with weight gain [49]. A cohort of British adults study also reported that red or processed meat consumption significantly predicted increased BMI and waist circumference [50]. Red and processed meats are part of the western dietary pattern, which is considered an obesity-inducing dietary pattern [51,52,53]. Red and processed meats contain high saturated fatty acid and cholesterol [54]. They are also categorized as high-energy-density food, which is directly associated with obesity [55,56]. Red meat intake is also positively related to some gut microbiota, which play a role in increasing obesity [57]. Furthermore, future studies should investigate why daily grilled foods consumption has a significant association with overweight and obesity, and why this relationship appears to be more prominent in rural areas. The type of foods and grilling methods used in urban and rural areas may be different and may explain the association with overweight and obesity.

Surprisingly, one factor that was found to have a protective effect against overweight and obesity was low educational attainment. Greater education may be associated with greater income and therefore greater access to high fat and high sugar snack foods. One earlier study also found an inverse relationship between educational level and overweight-obesity, agreeing with our finding [58]. The association between sociodemographic factors, including education level, depends on the development level of the country, wherein gender appeared to modify the effect [58,59]. Cohen et al. (2013) also suggests that the association between education and obesity may be affected by age [58]. As this study used data from a 10–18-year-old population, this may disguise the association between education and overweight/obesity as this population has roughly the same level of education. Moreover, formal educational attainment does not always reflect the level of knowledge in nutrition. It is possible that children and adolescents with lower educational attainment have more knowledge in nutrition, thus practicing a healthy diet and being more active, subsequently making them less likely to develop overweight and obesity. Previous studies have revealed that nutritional knowledge of either children or adolescents is related to overweight and obesity [60,61]. Parental factors such as knowledge, perception, and behavior might also play a role in childhood overweight and obesity [62,63].

There are some limitations to this study. First, the cross-sectional design prevents strong causal inferences. Second, no information was collected regarding total calories, macronutrients, and micronutrients from certain foods, thus disallowing us to examine whether the foods that children and adolescents consumed meet their daily nutritional needs. Third, the majority of data in this study were originally in categorical form, so that we were unable to transform them into a numeric scale.

Despite its limitations, our research provides suggestions for the development of effective prevention policies on the issue of the nutrition transition, which has been shown to increase the frequency of unhealthy foods consumption, obesity, and non-communicable diseases both in urban and rural areas. This study provides details on the types of food and food frequency, making it easier to recognize common foods that should be a concern for overweight and obesity in urban and rural areas. In addition, only a few questions about the foods were asked, and the help of food cards can reduce the bias in the interview process.

## 5. Conclusions

The current findings of the study identified the differences in overweight and obesity risk factors in rural and urban areas. Those living in urban areas had a higher percentage of overweight and obesity. Sedentary activity had a significant association with overweight and obesity not only in rural areas, but also in urban areas. However, dietary behavior has a different association with overweight and obesity risk in rural and urban areas.

Strategies to improve the nutritional status of children and adolescents in Indonesia should focus on developing approaches that vary by locale. Through understanding the risk factors both in rural and urban settings in Indonesia, these data can inform targeted community-based interventions for children and adolescent obesity prevention programs. In addition, our findings suggest that education, environmental, and policy interventions may need to specifically target urban settings, where access is high to a wide range of processed and traditional high-sugar, high-fat snack foods and beverages. Policies should be enacted that target access to these foods both within and immediately near school grounds. For example, sugar-sweetened beverages and caffeinated soft drinks and energy drinks should not be provided both at home and at school canteens in urban areas. Moreover, there is a need for policy-based strategies to limit the sugar-sweetened beverages and caffeinated soft drinks and energy drink exposure from the media. For both children and adolescents living in rural and urban areas, nutrition education should be integrated into the curriculum as a school-wide strategy that involves discussion on healthy eating and promotion of physical activity. This program should also involve parents so that healthy behaviors can continue after school.

In addition, in urban areas, sports centers could provide the opportunity for youth to join organized sports which are safe and gender friendly. Parks and public spaces should also enable children and adolescents in both genders to participate in various forms of physical activities. In rural settings where physical activity and sports facilities are lacking, we suggest a community approach to incorporating physical activity into everyday life in the natural environment (e.g., the use of walking trails, playing traditional games). The promotion of local-based foods that are high in fruits and vegetables and low in ultra-processed ingredients may also be advantageous.

## Figures and Tables

**Figure 1 nutrients-11-02813-f001:**
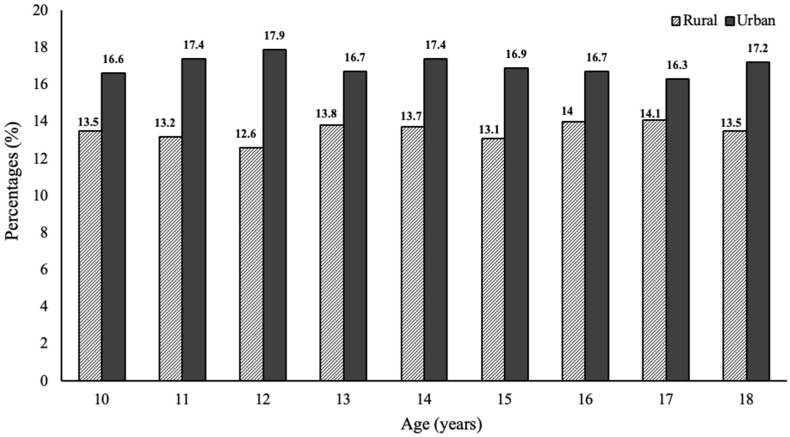
The percentage of overweight and obesity in rural and urban areas (%). *p*-value from *Chi-square* test was <0.0001 for each age, except 15–16 years old, where *p*-value was <0.05.

**Figure 2 nutrients-11-02813-f002:**
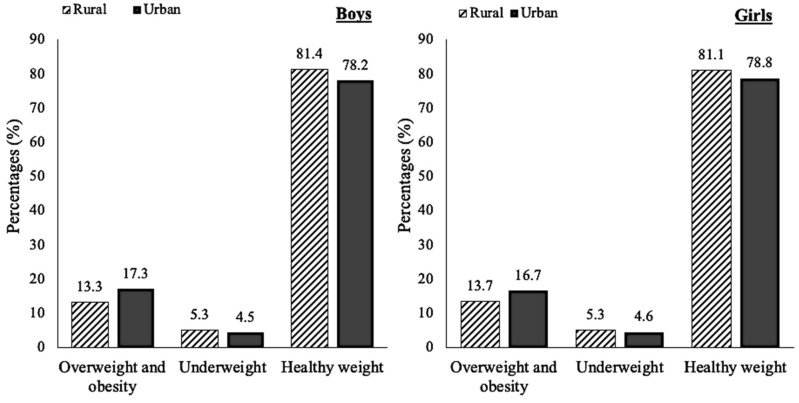
The percentage of overweight and obesity stratified by gender and locale. *p*-values from *chi-square* test for girls in urban and rural areas were <0.0001; also, boys in urban and rural areas were <0.0001.

**Figure 3 nutrients-11-02813-f003:**
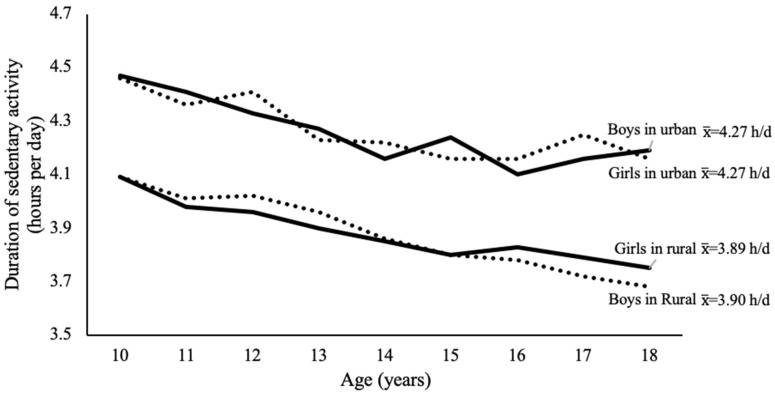
Duration of daily sedentary activity (hours/day) in each age among girls and boys in urban and rural areas. *p*-values from *t*-test for girls in urban and rural area, as well as boys in urban and rural areas, were <0.0001.

**Table 1 nutrients-11-02813-t001:** Multivariate analysis of overweight and obese risks.

Variables (Ref vs. Risk)		Overweight and Obese					
		Rural and Urban (*n* = 155,645)		Rural (*n* = 83,820)		Urban (*n* = 71,825)	
		OR	95% CI	OR	95% CI	OR	95% CI
Residency	Rural	ref					
	Urban	1.32 ***	1.28–1.35	-	-	-	-
Gender	Girls	ref		ref		ref	
	Boys	1.01	0.98–1.04	0.97	0.93–1.01	1.05 *	1.01–1.10
Age	15–18 years	ref		ref		ref	
	10–14 years	1.06 *	1.02–1.10	1.03	0.97–1.08	1.10 *	1.04–1.17
Education level							
High		ref		ref		ref	
Low (not completed in primary school)		0.92 *	0.89–0.96	0.93 *	0.87–0.98	0.91 *	0.86–0.97
Sedentary activity							
<2 h/day		ref		ref		ref	
2–4 h/day		1.12 ***	1.08–1.18	1.10 *	1.03–1.17	1.11 *	1.04–1.19
>4–6 h/day		1.09 ***	1.05–1.14	1.09 *	1.03–1.16	1.06	0.99–1.14
>6–8h/day		1.09 *	1.02–1.16	1.07	0.98–1.18	1.05	0.95–1.15
>8 h/day		1.15 ***	1.08–1.22	1.06	0.96–1.16	1.13 *	1.04–1.23
Sugar-sweetened beverages and foods							
<1 time/day		ref		ref		ref	
≥1 time/day		0.97	0.94–1.00	0.98	0.94–1.02	0.95 *	0.92–0.99
Coffee consumption	<1 time/day	ref		ref		ref	
	≥1 time/day	1.01	0.96–1.06	1.04	0.98–1.11	0.96	0.89–1.03
Caffeinated soft drinks and energy drinks consumption	<1 time/day	ref		ref		ref	
	≥1 time/day	1.11 *	1.03–.19	1.10	0.99–1.22	1.12 *	1.01–1.23
Fatty fried foods	<1 time/day	ref		ref		ref	
	≥1 time/day	1.00	0.96–1.03	0.96	0.92–1.01	1.03	0.99–1.07
Refined carbohydrate	<1 time/day	ref		ref		ref	
	≥1 time/day	1.01	0.97–1.05	0.98	0.92–1.03	1.04	0.99–1.09
Salty foods	<1 time/day	ref		ref		ref	
	≥1 time/day	1.03	0.99–1.07	1.09 *	1.04–1.15	0.98	0.93–1.03
Instant noodle	<1 time/day	ref		ref		ref	
	≥1 time/day	0.95 *	0.91–0.99	0.95	0.89–1.01	0.95	0.90–1.01
Preserved meats	<1 time/day	ref		ref		ref	
	≥1 time/day	0.95	0.89–1.01	0.93	0.83–1.03	0.97	0.89–1.06
Grilled foods	<1 time/day	ref		ref		ref	
	≥1 time/day	1.19 ***	1.12–1.26	1.32 *	1.22–1.42	1.03	0.94–1.14
Vegetables and fruits	≥5 portion/day	ref		ref		ref	
	<5 portion/day	0.92 *	0.88–0.98	0.96	0.89–1.04	0.90 *	0.83–0.96

Reference group: Sedentary activity: <2 h/day, dietary behavior: <1 time/day, * *p* < 0.05, *** *p*-value < 0.0001.

**Table 2 nutrients-11-02813-t002:** Percentages of dietary behaviors of children and adolescents stratified by locale and sex, *n* = 155,645.

Variables	Boys		*p*-Value	Girls		*p*-Value
	Rural (*n* = 43,207)	Urban (*n* = 36,219)		Rural (*n* = 40,613)	Urban (*n* = 35,606)	
**Sugar-sweetened beverages and foods**						
<1 time/day	51.2	44.0	<0.0001	53.2	46.1	<0.0001
≥1 time/day	48.8	56.0		46.8	53.9	
**Coffee consumption**						
<1 time/day	83.7	88.9	<0.0001	90.8	94.9	<0.0001
≥1 time/day	16.3	11.1		9.2	5.1	
**Caffeinated soft drinks and energy drinks consumption**						
<1 time/day	95.6	95.0	<0.0001	96.6	96.2	0.0011
≥1 time/day	4.4	5.0		3.4	3.8	
**Fatty fried foods**						
<1 time/day	70.3	62.0	<0.0001	69.3	61.2	<0.0001
≥1 time/day	29.7	38.0		30.7	38.8	
**Refined carbohydrate**						
<1 time/day	84.6	76.8	<0.0001	83.2	75.9	<0.0001
≥1 time/day	15.4	23.2		16.8	24.1	
**Salty foods**						
<1 time/day	78.9	79.1	0.5847	78.9	78.3	0.0536
≥1 time/day	21.1	20.9		21.1	21.7	
**Instant noodle**						
<1 time/day	85.5	85.2	0.3081	85.4	85.6	0.3937
≥1 time/day	14.5	14.8		14.6	14.4	
**Preserved meats**						
<1 time/day	96.0	93.8	<0.0001	95.9	93.5	<0.0001
≥1 time/day	4.0	6.2		4.1	6.5	
**Grilled foods**						
<1 time/day	92.8	95.1	<0.0001	93.1	94.9	<0.0001
≥1 time/day	7.2	4.9		6.9	5.1	
**Vegetables and fruits**						
≥5 portion/day	6.7	7.4	0.0001	6.8	7.0	0.2831
<5 portion/day	93.3	92.6		93.2	93.0	

Percentage (%) and *p*-value from chi square.

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
