# Peer review of "Rural–Urban Differences in Dietary Behavior and Obesity: Results of the Riskesdas Study in 10–18-Year-Old Indonesian Children and Adolescents"

_nutrients, 2019, doi:10.3390/nu11112813_

Round 1

Reviewer 1 Report

Some minor edits are needed:

Line 54: “Among children aged 5-12 years, the proportion of overweight remained unchanged 54 between 2013 and 2018 (10.8%), whereas the obesity prevalence was slightly increasing from 8.8% to 55 9.2% during these years.”. References are needed to support these estimates. Line 254: “A previous study reported that children and adolescents consuming any sweet drink or soft 254 drink (38).” The sentence seems incomplete. My previous comments: “Daily caffeinated beverages consumption was found to be associated with a higher OR in urban areas. What’s the mechanism behind? And related policy implications? “, Author response: Response: We thank the reviewer for this helpful suggestion and we made expanded our description of potential mechanisms and policy implications in line 277-279. However, I could not find the related edits regarding caffeinated beverages consumption in lines 277-279.

Author Response

Line 54: “Among children aged 5-12 years, the proportion of overweight remained unchanged 54 between 2013 and 2018 (10.8%), whereas the obesity prevalence was slightly increasing from 8.8% to 55 9.2% during these years.”. References are needed to support these estimates.

Response: We thank the reviewer for this helpful suggestion and have made additional references (Line 57):

National Institute of Health Research and Development. The Main Results of Riskesdas 2018. Jakarta: Ministry of Health, Republic of Indonesia. 2018. National Institute of Health Research and Development. Riset Kesehatan Dasar 2013. Jakarta: Ministry of Health, Republic of Indonesia. 2013.

Line 254: “A previous study reported that children and adolescents consuming any sweet drink or soft 254 drink (38).” The sentence seems incomplete.

Response: We thank the reviewer for this suggestion and made additional information in this sentence.

A previous study reported that children and adolescents consuming any sweet drink or soft drink was >70% and the proportion remained steady across time (line 364-365)

My previous comments: “Daily caffeinated beverages consumption was found to be associated with a higher OR in urban areas. What’s the mechanism behind? And related policy implications? “, Author response: Response: We thank the reviewer for this helpful suggestion and we made expanded our description of potential mechanisms and policy implications in line 277-279. However, I could not find the related edits regarding caffeinated beverages consumption in lines 277-279.

Response: We appreciate this helpful recommendation. We have modified the sentences and have made additional references “While caffeine is the key ingredient in most energy drinks, most products available on the market also contain a high amount of glucose or artificial sweetener  (41), which may later contribute to higher energy intake. Moreover, the consumption of energy drinks presumably leads to higher carbohydrate oxidation and lower lipid oxidation (42). Regarding soft drinks, besides a high level of added sugar, poor satiety and insufficient total energy compensation may influence weight gain (39). Finally, soft drink consumption can be used as an indicator of poor diet as it is linked to increased calorie intake and may displace nutritious food  (43). For regular caffeinated sugar-sweetened beverage consumption, caffeine content was found to be additive, thus promoting repeated consumption of sugar-sweetened beverages themselves, thus leading to weight gain (44). Children and adolescents in urban areas may be better able to find caffeinated beverages than in rural area. A policy to limit caffeinated beverages consumption in urban areas is needed, particularly targeting regulations in the advertisement, sale, and location of food retailers. School- and family-level supports are required to help children and adolescents make healthy food choices.”.

Reviewer 2 Report

The changes made by authors have improved the manuscript and its contributions to the literature

Author Response

Response: We thank the reviewer for their time in reading and critiquing this paper, and appreciate your positive comments.

This manuscript is a resubmission of an earlier submission. The following is a list of the peer review reports and author responses from that submission.

Round 1

Reviewer 1 Report

This article needs attention to methods and to the writing, as articulated by the reviews that went into detail abouit improvements.  However, it contains  some useful data that should be able to be of sufficient quality for public use.  

Improvements need to be made in the introduction, methods, and possibly results. In the introduction more attention needs to be made to the situatlin in Indonesia. In the methods more tdetail must be provided on measures, and rationale for analyses. In results consider transformed BMI scores or argue that they are not needed.

Author Response

Comments and Suggestions for Authors

This article needs attention to methods and to the writing, as articulated by the reviews that went into detail about improvements.  However, it contains some useful data that should be able to be of sufficient quality for public use. 

Improvements need to be made in the introduction, methods, and possible results. In the introduction, more attention needs to be made to the situation in Indonesia.

Response: We thank the reviewer for this helpful suggestion and have changed the introduction, methods, and results. We also made additional changes in the introduction section by adding the current prevalence of overweight and obesity prevalence in Indonesia inline 53-59.

In the methods, more detail must be provided on measures and rationale for analyses.

Response: We have provided more detail on measures, particularly additional description of dietary behaviour inline 114-130.

In results consider transformed BMI scores or argue that they are not needed.

For methods and results sections, we used BMI as a categorical variable because the majority of data in this study were originally in the categorical forms so that we were unable to transform them into numeric scale. Therefore, multiple logistic regression was performed to compare associated factors of obesity between urban and rural areas. We have added this as a limitation in the discussion inline 319-320.

Reviewer 2 Report

This study examined dietary intake and physical activity patterns and their relationship with weight status in Indonesian children and adolescents. This is an interesting study. However, there are a number of issues that need to be addressed:

Abstract

This is a cross-sectional study. Associations between some factors and obesity do not necessarily mean these are risk factors of obesity. The authors should be cautious about their wording. For example, “low physical activity was associated with overweight and obesity risk…”. Low physical activity may lead to obesity/overweight, and obesity may also lead to low physical activity. Casual inference could not be readily drawn from the estimates.  Also, it is unclear what confounds were controlled in the regression analyses. This also need to be reported in the abstract briefly.

Introduction

The authors talked about overweight and obesity in LMIC, for example, the prevalence of overweight and obesity in the developing world has been increasing. However, it is unclear how serious the problem is in Indonesia. Prevalence estimates/numbers, temporal trends and current status should be introduced and related reference should be cited,

Also, gaps in the literature and the significance of this study should be better explained.

Methods

Sample. Inclusion and exclusion criteria of the observations for analyses should be explicitly reported. Proportion of missing values, and how missings were handled should be explained.

The survey questionnaire asked “the frequency of 9 items of unhealthy foods consumption, including sugar-sweetened beverages and foods, coffee and caffeine-containing beverages,…” , but in the regression, there was a category named “Caffeinated soft drinks and energy drinks consumption”. It is unclear where that “energy drinks” were derived from and the reasoning for not using the original categories from the survey. Also, sugar-sweetened beverages did not include Caffeinated soft drinks? All these need better explanations/justifications.

Results

Figure 1. Why the prevalence estimates of rural were negative?

Figure 2. Either to use a histogram or a table, but not both.

Figures 1-3, statistical tests should be conducted and reported about the significance of the differences.

Table 2. It is unclear which groups were the reference groups for the variables in the logistic regression.

In addition to use overweight and obese as outcome, what about using BMI or BMI-z as outcome? Did the authors conduct the analyses? Were the results consistent? It would be more informative for the authors to report such estimates as well.

Discussion

Again, the authors should be cautious to male causal inference. For example, how to understand the relationship between physical activity and overweight/obese based on the estimates?

Daily caffeinated beverages consumption was found to be associated with a higher OR in urban areas. What’s the mechanism behind? And related policy implications?

Also, a negative relationship between educational level and overweight/obese in males was suggested by the regression estimates. Is this result robust to different model specifications? It’s not helpful by only saying “further study is needed to identify possible pathway”. The authors should provide some plausible explanations.

How should nutrition education be tailored appropriately to sociodemographics, physical activity, and residency based on these findings?

Author Response

Comments and Suggestions for Authors

This study examined dietary intake and physical activity patterns and their relationship with weight status in Indonesian children and adolescents. This is an interesting study.

Response: We thank the reviewer for their time in reading and critiquing this paper, and appreciate your positive comments.

However, there are a number of issues that need to be addressed:

Abstract

This is a cross-sectional study. Associations between some factors and obesity do not necessarily mean these are risk factors of obesity. The authors should be cautious about their wording. For example, “low physical activity was associated with overweight and obesity risk…”. Low physical activity may lead to obesity/overweight, and obesity may also lead to low physical activity. The causal inference could not be readily drawn from the estimates.  Also, it is unclear what confounds were controlled in the regression analyses. This also needs to be reported in the abstract briefly.

Response: We agree with the reviewer that this is confusing and have made the clarification about in line 27-28 of the revised manuscript. We changed the previous sentences to these new sentences “High duration of sedentary activity and high intake of unhealthy foods were associated with a high risk of overweight and obesity”. We have also checked the manuscript and made additional changes in lines 32-34. We controlled all variables, such as residency, gender, age, education level, physical activity, and food intake in the regression analysis.

Introduction

The authors talked about overweight and obesity in LMIC, for example, the prevalence of overweight and obesity in the developing world has been increasing. However, it is unclear how serious the problem is in Indonesia. Prevalence estimates/numbers, temporal trends and current status should be introduced and related reference should be cited,

Response: We agree with the reviewer that the problem of overweight and obesity among children and adolescents increasing dramatically should be described in the introduction. We have added sentences regarding the overweight and obesity problem in Indonesia in line 54-60.

Gaps in the literature and the significance of this study should be better explained.

Response: We added a sentence to clarify the gaps in the literature at the end of the introduction paragraph, line 73-75. We also have made additional sentences to expand the significance of this study in line 80-82.

Methods

Sample. Inclusion and exclusion criteria of the observations for analyses should be explicitly reported. The proportion of missing values, and how missings were handled should be explained.

Response: We thank the reviewer for this suggestion and made additional information regarding inclusion and exclusion criteria in line 94-97 and also have modified how missing data were handled line 98-100.

The survey questionnaire asked “the frequency of 9 items of unhealthy foods consumption, including sugar-sweetened beverages and foods, coffee and caffeine-containing beverages,…”, but in the regression, there was a category named “Caffeinated soft drinks and energy drinks consumption”. It is unclear where that “energy drinks” were derived from and the reasoning for not using the original categories from the survey. Also, sugar-sweetened beverages did not include Caffeinated soft drinks? All these need better explanations/justifications.

Response: We thank the reviewer for this helpful suggestion and have changed the terms of caffeine-containing beverages to caffeinated soft drinks and energy drinks. The description of caffeinated soft drinks and energy drinks was described in line 124-125. In addition, in this study, sugar-sweetened beverages without caffeine were not included in caffeinated soft drinks and energy drinks term line 121-123.

Results

Figure 1. Why the prevalence estimates of rural were negative?

Response: We appreciate this helpful recommendation and have changed the figure 1 without negative results. We made a new figure that easy to understand by readers. Previously, I want to show it in pyramid shape but the number should negative, then I decided to change the shape of the figure and also the percentage each age and prevent misleading when read overweight and obesity prevalence.

Figure 2. Either to use a histogram or a table, but not both.

Response: We thank the reviewer for this helpful suggestion and are just providing a table.

Figures 1-3, statistical tests should be conducted and reported about the significance of the differences.

Response: We thank the reviewer for this suggestion and have reported the significance of the differences in figures 1-3.

Table 2. It is unclear which groups were the reference groups for the variables in the logistic regression.

Response: We thank the reviewer for this suggestion and have added the reference groups.

In addition to using overweight and obese as an outcome, what about using BMI or BMI-z as the outcome? Did the authors conduct the analyses? Were the results consistent? It would be more informative for the authors to report such estimates as well.

We thank the reviewer for this suggestion and have reanalyzed it using linear regression and the results were consistent.

Discussion

Again, the authors should be cautious to make the causal inference. For example, how to understand the relationship between physical activity and overweight/obese based on the estimates?

Response: We appreciate this helpful recommendation. We changed the words “low physical activity to sedentary activity” to prevent misleading.  We also have changed the sentences which explain the reason boys tend to have a higher risk of obesity than girls (lines 258-260).

Daily caffeinated beverages consumption was found to be associated with a higher OR in urban areas. What’s the mechanism behind? And related policy implications?

Response: We thank the reviewer for this helpful suggestion and we made expanded our description of potential mechanisms and policy implications in line 277-279.

Also, a negative relationship between educational level and overweight/obese in males was suggested by the regression estimates. Is this result robust to different model specifications? It’s not helpful by only saying “further study is needed to identify possible pathway”. The authors should provide some plausible explanations.

Response: We agree with the reviewer and have changed these sentences. We have modified the sentences “greater education may be associated with greater income and therefore greater access to a high fat, high sugar snack foods”.

How should nutrition education be tailored appropriately to socio-demographics, physical activity, and residency based on these findings?

Response: We appreciate this helpful recommendation on how nutrition education is tailored. We have made addition sentences to explain the recommendation in line 316-329. Our findings suggest that education, environmental and policy interventions may need to specifically target urban settings, where access is high to a wide range of processed and traditional high sugar, high fat snack foods and beverages.  Policies should be enacted that target access to these foods both within and immediately near school grounds.